# VLAs Are Confined yet Capable of Generalizing to Novel Tasks

## Abstract

Vision-language-action models (VLAs) often achieve high performance on demonstrated tasks but struggle significantly when required to extrapolate, recombining skills used in different tasks in novel ways. For instance, VLAs might successfully put the cream cheese in the bowl and put the bowl on top of the cabinet, yet still fail to put the cream cheese on top of the cabinet. This motivates us to investigate whether VLAs merely overfit to demonstrated tasks or still hold the potential to extrapolate. Our study uses *text latent* as the ingredient; it is a task-specific vector derived from the models' hidden states. It thus encodes semantics necessary for completing a task and can be used to reconstruct the associated task behavior by writing it to the model's residual stream. Furthermore, we find that skills used in distinct tasks can be combined to produce novel behaviors by blending their respective *text latent*. Applying this to $\pi_0$, we increase its success rate from 9% to 83% on the proposed *libero-ood* benchmark, which features 20 tasks extrapolated from standard LIBERO tasks. This reveals that the skill representations encoded in *text-latent* are individual yet composable, while $\pi_0$ fails to autonomously combine these representations for extrapolation. This also validates the design of *libero-ood*; it comprises tasks that the model fails, yet should be able to complete. We then tested other VLAs on *libero-ood*, and none of them achieved a success rate higher than 21%. Further analysis reveals VLAs share a common pattern to exhibit spatial overfitting, associating object names with where the object is spatially located in the demonstrated scene rather than achieving true object and goal understanding.

## 1 Introduction

Building toward generalist, vision-language-action models (VLAs) trained with large-scale multimodal datasets (O'Neill et al., 2023) have shown remarkable visual and language generalizability, leading to strong performance on diverse manipulation tasks (Brohan et al., 2023; O'Neill et al., 2023; Li et al., 2023b; Kim et al., 2024; Durante et al., 2024; Huang et al., 2024; et al., 2024; Zhen et al., 2024; Black et al., 2024; Team et al., 2024; Hou et al., 2025; Qu et al., 2025; Zheng et al., 2024). Typically, for satisfactory deployment on new tasks, VLAs are fine-tuned using demonstrations of target tasks (Kim et al., 2025). Though this paradigm ensures good in-distribution generalization to light conditions (Li et al., 2024b) and small perturbations of scene layout (Liu et al., 2024), we empirically find that VLAs struggle with out-of-distribution (OOD) tasks, particularly those extrapolated from tasks that they can perform well individually. For instance, a VLA might successfully "put the cream cheese in the bowl" and "put the bowl on top of the cabinet", yet fail to perform the extrapolated task of "put the cream cheese on top of the cabinet", even though the motion primitives required by this task, "picking the cream cheese" and "reaching the top of the cabinet", have already been learned in different tasks. This raises a question: **Do VLAs merely overfit to demonstrated trajectories or do they learn composable representations that support broader generalization?** We study this on the SOTA VLA $\pi_0$ (Black et al., 2024) with its *text latent*, which is a vector derived from the model's internal states, and thus encodes the learned representations.

To identify the *text latent* for a given task, we run $\pi_0$ on the corresponding task demonstrations and record the hidden states of text tokens for each transformer layer. After this, we average all collected layer-wise features and obtain *text latent*. By writing the *text latent* to the text tokens' residual streams (Elhage et al., 2021) of $\pi_0$, we can reconstruct the associated task trajectory without

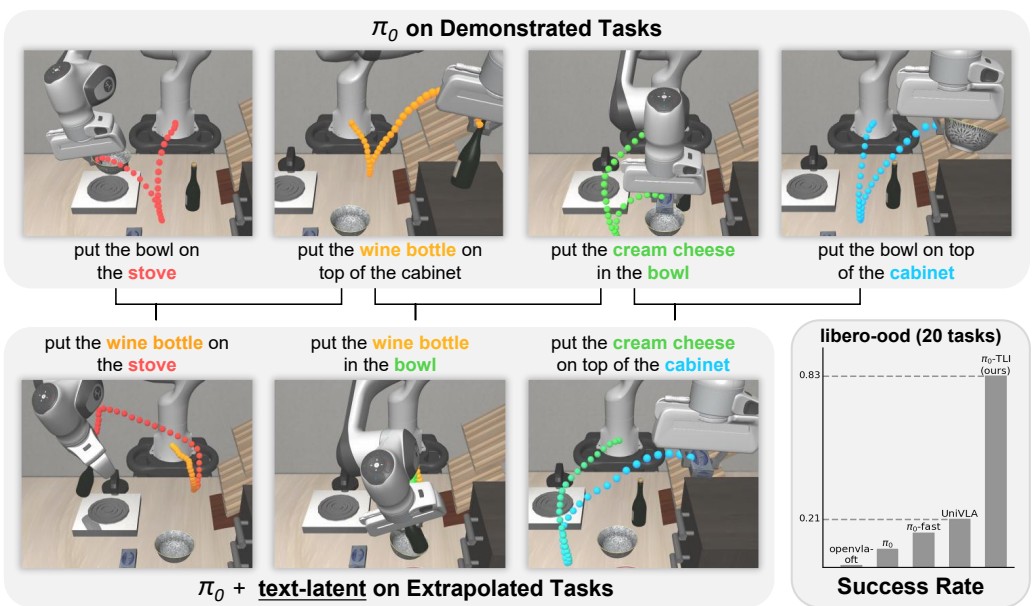

Figure 1: UniVLA, openvla-oft, $\pi_0$, and $\pi_0$-fast achieve less than 21% success rate on the proposed *libero-ood* benchmark. It consists of tasks extrapolated from standard LIBERO (Liu et al., 2024) tasks that those VLAs have been fine-tuned with. By applying the proposed *Text Latent Interpolation* (TLI), we improve the performance of $\pi_0$ up to 83%. Behaviors of $\pi_0$-TLI on three exemplary extrapolated tasks are shown in the second row, while the first row shows the associated two base tasks.

providing the task prompt. We also find that unembedding *text latent* (nostalgebraist, 2020) produces alternative task prompts, which can instruct the model to finish the corresponding tasks with about 70% success rate. However, these alternative prompts are encrypted and unreadable, enabling private instruction and backdoor attacks. Furthermore, we find that by using *Text Latent Interpolation* (TLI) to blend the representations learned for two tasks, the model can combine respective sub-behaviors or skills to complete novel tasks. Specifically, TLI injects the temporal interpolation of the two respective *text latents* to the residual stream, with the mixing ratio linearly adjusted at each timestep. As a result, the intervention guides the model towards Task 1's behavior at the beginning, and as the episode progresses, the influence smoothly shifts towards Task 2's behavior. As shown in Fig. 1, $\pi_0$ with *text latent* can stitch together sub-trajectories used in different tasks to finish extrapolated tasks, even if the newly composed trajectories are not shown in training and fine-tuning data.

To verify that learned representations can broadly support OOD generalization with TLI, we introduce the *libero-ood* task suite. It comprises 20 challenging extrapolated tasks derived from the three standard LIBERO suites: *libero-goal*, *libero-spatial*, and *libero-object*. Each task in *libero-ood* is designed such that while the individual movements required for grasping and placement are present in separate training tasks, the specific combination of these movements is novel. By applying TLI, the success rate of $\pi_0$ on *libero-ood* increases from 9% to 83%. And it manages to complete 19 out of 20 tasks at least once in 10 repeated runs. **This significant improvement confirms that the skill representations, encoded in *text latents*, generally exist and are composable, while the model cannot combine them autonomously and hence failed on novel extrapolated tasks.** On the other hand, this result validates the design of *libero-ood* as a suitable benchmark for OOD generalizability for all VLAs. Rather than presenting VLAs in completely impossible OOD tasks (e.g., holding a steering wheel to drive a car), *libero-ood* comprises OOD tasks that the model has the potential to solve by simply mixing the learned representation. Therefore, we tested other SOTA VLAs on *libero-ood*, including $\pi_0$-fast (Pertsch et al., 2025), openvla-oft (Kim et al., 2025), and UniVLA (Bu et al., 2025). Though they achieve about a 95% success rate on the three standard LIBERO task suites after fine-tuning, their success rate on *libero-ood* is less than 21%, highlighting their limitations in extrapolation. Further qualitative analysis reveals that VLAs commonly exhibit spatial overfitting, associating object names with their locations in the demonstrated scene. When instructed to pick an object, they always move to where the object has been placed before, ignoring its current location. It uncovers that VLAs fail to learn true object or goal understanding even after fine-tuning.

## 2 RELATED WORK

**Vision-Language-Action Models.** VLAs are usually initialized from vision-language models (VLMs) pretrained with large-scale cross-modality data. To adapt it for decision-making, we need to further train it with large-scale cross-embodiment datasets like Open X-Eombodiment (O'Neill et al., 2023), following a fine-tuning on specific tasks with fewer demonstrations (Brohan et al., 2023; O'Neill et al., 2023; Li et al., 2023b; Kim et al., 2024; Durante et al., 2024; Huang et al., 2024; et al., 2024; Zhen et al., 2024; Black et al., 2024; Team et al., 2024; Hou et al., 2025; Bu et al., 2025; Qu et al., 2025; Wang et al., 2024; NVIDIA et al., 2025; Team et al., 2025; Pertsch et al., 2025; Zheng et al., 2024). Though VLAs are all built upon the transformer-based pre-trained VLMs (Vaswani et al., 2023), it remains unclear what is the best way to decode actions. OpenVLA (Kim et al., 2024), RT-2 (Brohan et al., 2023), and $\pi_0$-fast (Pertsch et al., 2025) follow the next-token prediction manner used by VLMs and LLMs and decode discrete action tokens, which will be converted to continuous actions according to different tokenization schemes. Another widely adopted way to predict action is to use a regression head or diffusion model with extra parameters (NVIDIA et al., 2025; Black et al., 2024; Kim et al., 2024; Wen et al., 2024; Lee et al., 2024; Chi et al., 2023; Kim et al., 2025; Li et al., 2024a). As the VLM is the core component for VLAs' ability, we choose to mine the meaningful internal representation of the transformer part of $\pi_0$, which adopts an additional action expert to generate continuous actions with flow matching.

**Mechanistic Interpretability (MI).** This is a subfield of interpretability, where researchers reverse engineer the model's internal computations to understand how it works (Rai et al., 2025; Elhage et al., 2021; Vaswani et al., 2023) and reconstruct or activate certain behaviors. Some early works analyze image classification models (Olah et al., 2020; Bau et al., 2017; Zhou et al., 2015) and find that neurons play the role of feature detectors for patterns from simple curves to complex objects like cats (Cammarata et al., 2020). Recent studies of MI on LLM and VLM identifies important circuits that can perform specific tasks like induction head (Olsson et al., 2022) to output repeating words, function vectors (Todd et al., 2024; Luo et al., 2024) to produce antonyms, attention head to detect number or shape (Gandelsman et al., 2024), neurons contributing to recognize specific objects (Gandelsman et al., 2024), and internal features making VLMs hallucinate (Jiang et al., 2025). However, there are limited works that study the neural policies or VLAs from the MI perspective. The most relevant works study adversarial attack with feature attribution (Wang et al., 2025), symbolic representation uncovering (Lu et al., 2025), and motion-relevant neuron identifying and characterization (Li et al., 2023a; Häon et al., 2025). Our work finds causal effects between the internal representations and VLAs' behaviors. As a result, we can produce obscure prompts with *logit lens* (nostalgebraist, 2020), enabling private instruction or backdoor attack. In addition, the identified functional component enables the $\pi_0$ for extrapolated tasks, where all SOTA VLAs struggle with.

## 3 METHOD

We start by formulating how transformer-based VLAs work. Then, we illustrate how to identify *text latent* and how to use them to change the model's internal representation, steering its behavior.

### 3.1 PRELIMINARY

Existing VLAs adopt VLMs as encoders to fuse both vision and language information. Concretely, a pre-trained vision encoder, e.g., CLIP (Radford et al., 2021) and SigLIP (Zhai et al., 2023), is used to generate $d$-dimensional sequential image embeddings from image patches, followed by $d$-dimensional text embeddings that are tokenized and projected from the task description. For some VLAs (Kim et al., 2024), the task description is encapsulated with certain context, like "what actions the robot should take to {*task description*}", which introduces extra tokens. In addition, the proprioceptive state will be projected (Kim et al., 2024) or tokenized (Pertsch et al., 2025) into the $d$-dimensional shared space to work with image-text embeddings. After tokenization, we assume there are totally $m$ $d$-dimensional embeddings, $e = \{e^i : e^i \in \mathbb{R}^d, i = 1...m\}$, obtained from image, text, and robot proprioception, which will go through $L$ transformer layers. Except for the last layer, each layer $l$ outputs hidden states $h_l = \{h_l^i : h_l^i \in \mathbb{R}^d, i = 1...m\}$. The action is then generated by $a = f(e, h)$, where $h = \{h_l : h_l \in \mathbb{R}^{m \times d}, l = 1...L-1\}$ is hidden states of all tokens across $L-1$ layers. More precisely, the action generation uses the KV cache derived from $e$ and $h$, where the image-text

embeddings $e$ are passed through the first transformer layer (layer 0) to compute key and value projections, and the hidden states $h_l$ are passed through their subsequent layer $l + 1$ to compute the corresponding KV projections. To simplify the notation, we condition the action generation on $e$ and $h$. This formulation applies to most existing VLAs as long as the VLM is reused to do action generation in an autoregressive way (Kim et al., 2024; Qu et al., 2025; Brohan et al., 2023; Pertsch et al., 2025) or the extra action prediction module is also transformer-based (NVIDIA et al., 2025; Black et al., 2024), which can take the KV-cache to do causal self-attention.

## 3.2 TEXT LATENT

As VLAs' behavior depends on the task description, we hypothesize that the skill representations are embedded in the internal representations of text (task description) tokens. To study this mechanism, we denote the text embeddings as $e^T = \{e^i : e^i \in \mathbb{R}^d, i \in T\}$ and their hidden state at each layer as $h_l^T = \{h_l^i : h_l^i \in \mathbb{R}^d, i \in T\}$, where $T$ is the set of text tokens indices. Therefore, the hidden states of all text tokens across all $L - 1$ layers can be represented by a tensor as $h^T \in \mathbb{R}^{L-1 \times |T| \times d}$ after preserving orders and stacking them. By denoting the rest of the embeddings and their hidden states as $e^- = e \setminus e^T$ and $h^- = h \setminus h^T$, we can update the action generation function at timestep $i$ as

$$a = f(e^T, h^T(i), e^-(i), h^-(i)) \tag{1}$$

Note that text embedding, $e^T$, doesn't condition on $i$, since it is fixed throughout the whole episode, while the rest image and proprioceptive tokens are changed at different timesteps. Our goal is to manipulate the $h^T(i)$ at each decision-making step to activate behaviors with specific semantics.

*Text latent* is the ingredient for doing this. It has the same shape as $h^T$ and thus can be written into the text tokens' residual stream. We identify it by averaging a set of text hidden states $\{h^T(i) : h^T(i) \in \mathbb{R}^{L-1 \times n \times d}, i \in B\}$, where $B$ is all timestep indices of a demonstrated episode for the target task, and $h^T(i)$ is thus obtained by forwarding the model with the observation at timestep $i$. As multiple demonstrated episodes exist for a single task in the training sets, the average is thus taken over $K$ demonstrations. Therefore, the *text latent* can be calculated by element-wise average:

$$\mathcal{T} = \frac{1}{\sum_{k=1}^{k=K} |B_k|} \sum_{k=1}^{k=K} \sum_{i \in B_k} h^T(i) \tag{2}$$

In our reconstruction experiment, we demonstrate that $\mathcal{T}$ captures the most essential knowledge for finishing the corresponding task. And blending them enables skill combination for task extrapolation.

## 3.3 TEXT LATENT INTERPOLATION

A task derived from two base tasks can be interpreted as starting with Task 1 and gradually switching to Task 2. Though text token ids are discrete and cannot be changed smoothly throughout the episode, we can approximate a continuous transition by linearly interpolating the text embeddings of the two base task prompts, $e_1^T$ and $e_2^T$, at timestep $i$. This is called *Text Embedding Interpolation* (TEI).

$$e^T = e^T(i) = (1 - \alpha)e_1^T + \alpha \, e_2^T, \quad \alpha = i/\lambda, \quad 0 \le i \le \lambda, \tag{3}$$

where the hyperparameter $\lambda$ controls the transition speed. The $\lambda$ is set to 24, the average number of policy-execution steps, for $\pi_0$, except for *put the wine bottle in the bowl*. As the wine bottle is near the bowl, we shorten $\lambda$ to 14 for this task, so the second task behavior is activated sooner.

Intuitively, TEI rewrites the task prompt at every step with a weighted blend of the two source instructions. Additionally, we can leave the target task prompt (and thus its embedding) unchanged and operate on the model's residual stream with the two respective *text latent* $(\mathcal{T}^1, \mathcal{T}^2)$. We term this approach *Text Latent Interpolation* (TLI), which modifies the text hidden states $h^T(i)$ as follows:

$$h^T(i) = h^T(i) + \left[(1 - \alpha)\,\mathcal{T}^1 + \alpha\mathcal{T}^2\right] - \left[(1 - \alpha)\,\mathcal{T}^2 + \alpha\mathcal{T}^1\right], \quad \alpha = i/\lambda, \quad 0 \le i \le \lambda. \tag{4}$$

At the beginning of the episode, Task 2's context is suppressed and subtracted from the residual stream. As the episode progresses ($\alpha : 0 \to 1$), Task 2's context is gradually injected into the residual stream while Task 1's context fades out and is finally suppressed. TLI can also be applied to a specific layer $l$ by replacing $h_l^T(i)$ with the interpolated pair $(\mathcal{T}_l^1, \mathcal{T}_l^2)$ in the same fashion. The full interpolation procedure is listed in Algorithm 1. Both TEI and TLI can be deployed independently or jointly. In either case, the ratio $i/\lambda$ is clipped between 0 and 1. If the length of the text dimension of $\mathcal{T}^1$ or $\mathcal{T}^2$ differs from the number of tokens of the target prompt, we truncate it or pad it with zeros to match the length of the target prompt, as the task descriptions are of a similar length.

---

**Algorithm 1** TEI and TLI during Task Execution

---

**Require:** *text latent* $\mathcal{T}^1$, $\mathcal{T}^2$, Interpolation steps $\lambda$, Initial observations $o$, Max timestep $J$

1: **for** $i = 1$ to $J$ **do**
2:      $e^-, h^-, e^T, h^T \leftarrow \text{encode}(o)$        ▷ Get internal representation
3:      **if** *Text Embedding Interpolation* (TEI)
4:         $e^T = (1 - i/\lambda)\, e_1^T + (i/\lambda)e_2^T$        ▷ Overwrite text embedding
6:      **if** *Text Latent Interpolation* (TLI)
7:         $h^T(i) = h^T(i) + (1 - i/\lambda)(\mathcal{T}^1 - \mathcal{T}^2) + i/\lambda(\mathcal{T}^2 - \mathcal{T}^1)$    ▷ Write to residual stream
9:      $a = f(e^T, h^T(i), e^-(i), h^-(i))$        ▷ Decode action
10:      $o \leftarrow \text{simulation}(a)$        ▷ Forward simulation
11: **end for**

---

## 4 EXPERIMENTS

**Benchmark.** We conducted experiments with LIBERO (Liu et al., 2024), a simulation environment widely employed for evaluating Vision-Language-Action (VLA) models (Black et al., 2024; Pertsch et al., 2025; Kim et al., 2025; Qu et al., 2025; Zheng et al., 2024; Bu et al., 2025). We use three standard task suites from LIBERO: *libero-goal*, *libero-object*, and *libero-spatial*. Each suite contains 10 tasks, and most of them require pick-and-place. It is recommended to learn their details first in Appendix B. We also introduce a novel task suite, *libero-ood*, comprising two sub-suites *libero-goal-ood* and *libero-spatial-ood*. Each also contains 10 extrapolated tasks. The key idea behind these 20 tasks is to ensure that both the grasping and placement locations have individually appeared in the training data, so the policy has already learned how to reach these locations. However, the specific trajectory connecting these two locations has not been demonstrated. Thus, solving these tasks requires the policy to stitch together sub-trajectories it has learned from demonstrated tasks.

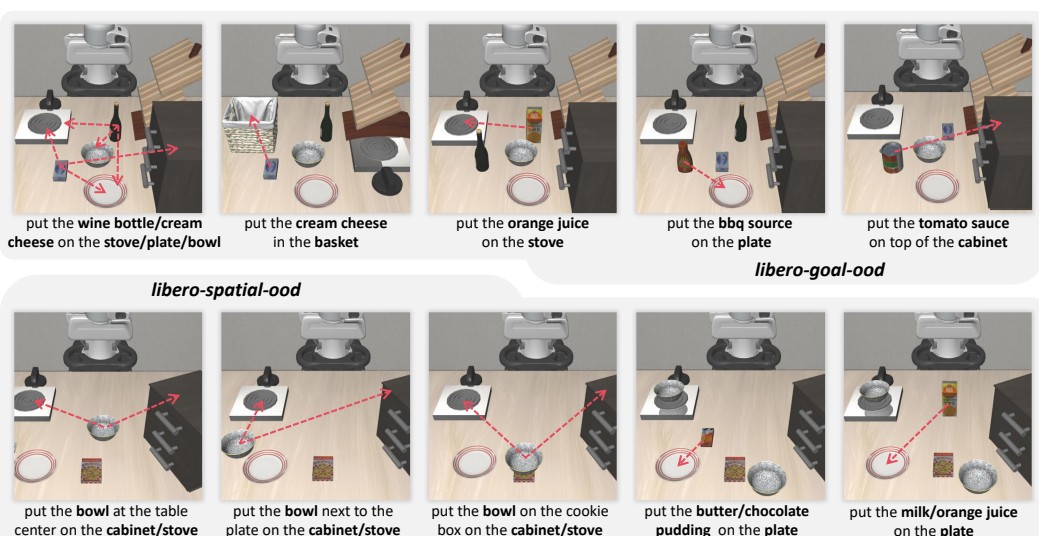

Figure 2: Visualization of scene layouts, object to pick and where to place (denoted by the red arrows), and prompts for tasks in *libero-ood*, which can be further split into *libero-goal-ood* and *libero-spatial-ood*. *libero-goal-ood* includes six extrapolated tasks that require combining sub-skills learned from *libero-goal* and operating in the same scene layout as *libero-goal* tasks (the top-left figure), while the remaining four tasks slightly change the layout and additionally demand transferring knowledge about objects. On the other hand, *libero-spatial-ood* provides six tasks that require transferring the skills learned in *libero-goal* and *libero-spatial* to place the object on the cabinet or stove. The remaining four tasks require placing unseen objects from *libero-object* at new locations on the plate.

**Experiments.** Our main experimental goal is to examine whether $\pi_0$ learns individual yet composable task representations, so it can complete these 20 extrapolated tasks by combining learned representations, encoded in *text latent*. For each task in the three standard suites, we identify the corresponding *text latent* using 20 demonstrations, calculated according to equation 2. We first demonstrate on $\pi_0$ that *text latent* encapsulates the essential knowledge required to complete standard LIBERO tasks by reconstructing the task behavior without clear task prompts. Second, we show that by interpolating between *text latent* or *text embeddings*, $\pi_0$ can complete challenging extrapolated tasks in *libero-ood* suite, where tasks pose difficulties for current state-of-the-art VLAs. Finally, our analysis across specific tasks reveals that VLAs' behavior relies on the mechanistic memorization of demonstrated trajectories, rather than a genuine conceptual understanding of objects or goals.

**Evaluation.** For each task, we execute 10 independent runs using different random seeds. Therefore, the final success rate for each task suite is calculated as the proportion of successful episodes across the total 100 runs (10 tasks × 10 runs/task). All experiments are conducted with Nvidia RTX 4090.

## 4.1 TASK RECONSTRUCTION

In this experiment, we use *text latent* to reconstruct the behavior or trajectory learned for the three standard task suites, demonstrating that individual task representation is encoded in *text latent*. We use two ways to examine the effectiveness of *text latent*. The first way is to mask all the text tokens or set each text token to the space character (" ") so the text prompt doesn't provide any task information. We can then add back the *text latent* to each hidden layer representation when executing the policy. The second way is to unembed the early layer vectors of *text latent* into a set of token ids (nostalgebraist, 2020), producing alternative prompts. Thus, we can feed the alternative prompt to the model directly.

|  | libero-object | libero-goal | libero-spatial |
|---|---|---|---|
| *Original* | *0.98* | *0.95* | *0.97* |
| Mask Prompt | 0.11 | 0.14 | 0.24 |
| Blank Prompt | 0.28 | 0.16 | 0.23 |
| Blank Prompt+$\mathcal{T}$ | **0.94** | **0.82** | 0.81 |
| $\mathcal{T}_1$-Prompt | 0.92 | 0.66 | **0.98** |
| $\mathcal{T}_2$-Prompt | 0.73 | 0.19 | 0.85 |
| $\mathcal{T}_3$-Prompt | 0.56 | - | 0.68 |

Table 1: The success rate of different ways to reconstruct tasks.

The results are shown in Table 1. The first line is the official performance of $\pi_0$ with the original task prompt, serving as the upper bound. The *Mask Prompt* experiment uses only image input for all task execution, while the *Blank Prompt* adds back text tokens but fills with space characters (" "). In both settings, the policy has no task instructions, indicating the lower bound of performance. The rest lines show the performance of different ways to reconstruct the task. For each task suite, the settings yielding the best performance are bolded, and the second-best performance is underlined. The experiment *Blank Prompt+$\mathcal{T}$* shows that by writing the *text latent* to the model's residual stream, $h^T(i) = h^T(i) + \mathcal{T}$, the tasks can be finished with a success rate higher than 80%, even if the prompt is blanked and provides none of the task information. **Thus, it confirms that *text latent* captures essential task knowledge, and can remind $\pi_0$ of a task by injecting it into the model's internal states.**

As $\mathcal{T} \in \mathbb{R}^{L-1 \times n \times d}$, we can thus applying the language embedding matrix $E$ to unembed $\mathcal{T}_l$ into tokens by calculating the cosine similarity between each column of $E$ and $\mathcal{T}_l$, then selecting the indices with maximum similarity as new tokens, composing the alternative prompt. As shown in Table 1, we unembed $\mathcal{T}_1$, $\mathcal{T}_2$, and $\mathcal{T}_3$ and find that for *libero-goal*, the prompt produced by unembedding $\mathcal{T}_1$ achieves a 66% success rate, while the next layer's vector can not reconstruct the task well. However, for *libero-object* and *libero-spatial*, even the prompt unembedded by $\mathcal{T}_2$ and $\mathcal{T}_3$ can reconstruct tasks with an average success rate of 70%. **We find most prompts unembedded from *text latent* are encrypted and unreadable, even for people who are familiar with the original prompts.** This enables an application, obscure prompting, for private instruction or backdoor attacks. The unembedded prompt for the three task suites is shown in Appendix C.

## 4.2 TASK EXTRAPOLATION

As *text latent* encodes task context or skill representations, we then ask whether the behaviors learned from separate tasks can be recombined using the respective *text latent*. We refer to any task that demands such a behavior recombination as an *extrapolated task*. These new tasks keep objects'

|  | UniVLA | openvla-oft | $\pi_0$-fast | $\pi_0$ | $\pi_0^S$ | $\pi_0$-TLI | $\pi_0$-TLI$^+$ | $\pi_0$-TLI* |
|---|---|---|---|---|---|---|---|---|
| libero-goal-ood | 0.32 | 0.01 | 0.13 | 0.02 | 0.72 | **0.85** | **0.85** | 0.33 |
| libero-spatial-ood | 0.11 | 0 | 0.17 | 0.16 | 0.66 | **0.81** | 0.69 | 0.18 |
| All | 0.21 | 0.01 | 0.15 | 0.09 | 0.69 | **0.83** | 0.77 | 0.25 |

Table 2: The success rate of SOTA VLAs and our methods on the *libero-goal-ood* task suite. Our methods enable $\pi_0$ for task extrapolation by simply combining its learned representation. Detailed performance of each task and the behavior visualization of $\pi_0$-TLI is available in Appendix E and D.

locations or layouts the same as the scenes for collecting training demonstrations. Thus, for each *libero-ood* task, VLAs have learned to reach the grasping and placement locations, respectively. However, $\pi_0$ only shows a 9% success rate in *libero-ood*. After applying the proposed TLI for $\pi_0$, we can largely improve its success rate to 83%, as shown in Fig. 3. **This confirms that skill representations generally exist in $\pi_0$ that can be rearranged to solve novel tasks. It also validates that the tasks in *libero-ood* are within $\pi_0$'s capability, while it fails on it.** We further test other SOTA VLAs' extrapolation ability using *libero-ood*. The results are shown in Table 2. None of the VLAs achieves a success rate higher than 21%. Among all VLAs, the best is UniVLA. It shows trajectory stitching behavior in some tasks, and better language following ability, which results from the disentangled training of the vision-language module and the action module (Huang et al., 2025).

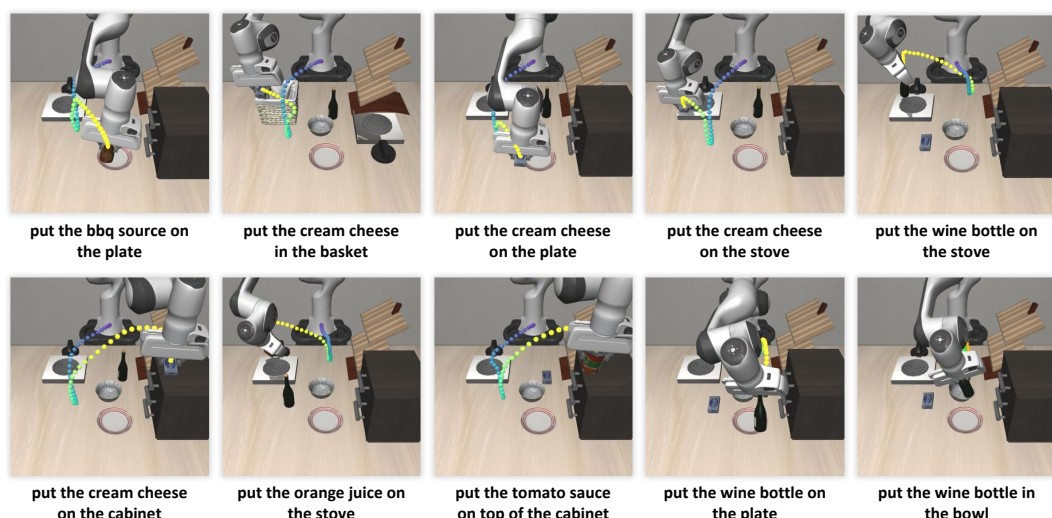

put the bbq source on the plate  put the cream cheese in the basket  put the cream cheese on the plate  put the cream cheese on the stove  put the wine bottle on the stove

put the cream cheese on the cabinet  put the orange juice on the stove  put the tomato sauce on top of the cabinet  put the wine bottle on the plate  put the wine bottle in the bowl

Figure 3: Visualization of $\pi_0$-TLI's behavior in *libero-ood*. Result on *libero-spatial* is in Appendix D.

We also applied both TEI and TLI to the model inference together, yielding $\pi_0$-TLI$^+$. The results show that TEI doesn't further improve the performance of $\pi_0$-TLI for *libero-goal-ood*, while it even slightly harms the performance of *libero-spatial-ood*. We will use $\pi_0$-TLI$^+$ to reveal the spatial overfitting exhibited in $\pi_0$ in the next section. In addition, we run another experiment ($\pi_0$-TLI*) using blank prompts similar to the previous reconstruction experiment. Its performance drastically drops to 33% and 18%, indicating the importance of the prompt for extrapolated tasks. The text embedding of an extrapolated prompt can be viewed as stitching two base task prompts $e^T = \mathbf{concat}(e_1^T[:a], e_2^T[b:])$, where $a$ and $b$ indicate the prompt truncation position. These stitched text prompts ($\pi_0$-TLI) are better for combining two base task behaviors than interpolated prompts ($\pi_0$-TLI$^+$).

On the other hand, $\pi_0$-TLI* can be viewed as implicitly switching task context without explicit prompts. This inspires us to try explicitly prompt switching to solve *libero-ood*. Concretely, we can feed Task 1's prompt to $\pi_0$, and when timestep $> \lambda/2$, we switch to Task 2's prompt. The $\pi_0^S$ executes in this way and reaches a 69% success rate. **It suggests that augmenting the model's ability with learned representation ($\pi_0$-TLI) is better than cheating it with what task it currently performs.** We also apply prompt switching to other VLAs, and openvla-oft, UniVLA, openpi-fast achieve 0%, 2%, 35% success rate, respectively, further highlighting the challenge of *libero-ood*.

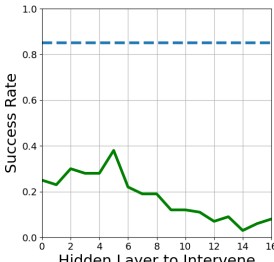

In addition, we try to find a more compact structure for *text latent*. Specifically, we intervene in a specific layer $l$ of the model with $\mathcal{T}_l$ to find whether it contributes significantly to the extrapolated tasks. As shown in the left figure, the early several layers can work alone to finish tasks in *libero-ood* with more than 20% success rate. After layer 6, the success rate begins to drop, while at layer 16 the success rate recovers to 10%. **As each layer of $\mathcal{T}$ can work independently and contributes more or less to the success, we decide to keep all of them.** As a result, the $\pi_0$ with all hidden layer interventions can achieve an 83% success rate indicated by the blue dashed line.

## 4.3 SPATIAL OVERFITTING OF $\pi_0$

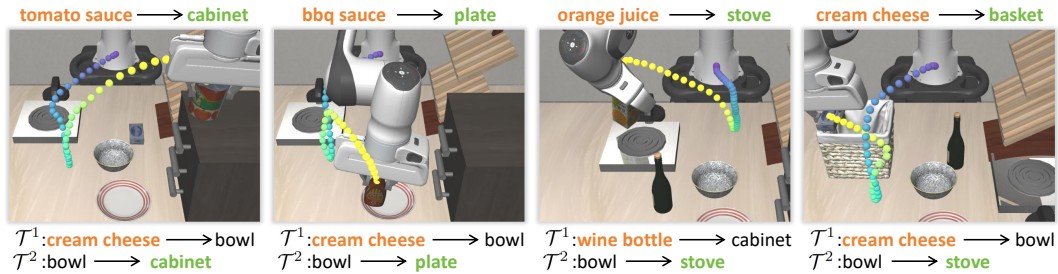

Figure 4: Trajectories of $\pi_0$-TLI$^+$ for the four tasks that require object recognition. Object to grasp and place to drop are shown on top of the figure; The two *text latent* used to complete the task are listed underneath. These behaviors reveal spatial overfitting. For example, we can use the *text latent* and text embedding of "placing on the stove" to put any object into the basket that occupied the original location of the stove in the training data, ignoring the current location of the stove.

As shown in Fig. 2, the rightmost 4 tasks in *libero-goal-ood* require the model to pick up objects that have never been shown in the scene of *libero-goal* before. In these 4 tasks, we swap the original objects with new ones from *libero-object*, while still keeping the displaced objects in the scene. Using both TEI and TLI, $\pi_0$ achieves an average success rate of 85% on these four tasks, which confirms the existence of *Spatial Overfitting* in $\pi_0$. Specifically, consider the task *put the orange juice on the stove*. The orange juice is placed at the original position of the wine bottle, and the wine bottle is moved to the former position of the cream cheese. We can instruct $\pi_0$ to pick the orange juice, using the *text latents* and text embeddings of *put the wine bottle on top of the cabinet* with $\pi_0$-TLI$^+$. Both text embedding and hidden states request $\pi_0$ to pick the wine bottle, while $\pi_0$ ignores the current location of the wine bottle that is reachable at the location of the cream cheese, and it still mechanically moves to where the wine bottle is placed in the demonstrated scene. The trajectories produced by $\pi_0$-TLI$^+$ for all four tasks are shown in Fig. 4. **This behavior indicates that the $\pi_0$ does not understand object identity but instead maps object names to fixed locations, termed as *Spatial Overfitting*.** Thus, "cream cheese" actually means "the object at the location where the cream cheese appeared during training." And $\pi_0$ can grasp anything placed at the location of the cream cheese.

## 4.4 SPATIAL OVERFITTING IS COMMON IN VLAS

The spatial overfitting commonly exists for VLAs besides $\pi_0$, which can be confirmed by experiments on *libero-object* suite, where five tasks place the target object at the center of the scene, and the other five place it in the top-right corner. After fine-tuning with demonstrations, VLAs can always pick the central object with any of the five "center" prompts and the top-right object with any of the five "top-right" prompts. In the two-prompt experiments shown in Table 3, we always use the prompt, *pick up the cream cheese* to pick the object at the scene center, and the prompt, *pick up the alphabet soup* for the objects at the top-right corner. We then run the *libero-object* with the two prompts. All SOTA VLAs can maintain the performance on the *libero-object* with incorrect prompts, even though the "cream cheese" and "alphabet soup" still appear in the scene somewhere.

|  | $\pi_0$ | $\pi_0$-fast | openvla-oft | UniVLA |
|---|---|---|---|---|
| Two-prompt | 0.94 | 0.86 | 0.99 | 0.90 |
| OOD-position | 0 | 0 | 0 | 0 |

Table 3: The success rate on modified *libero-object* suite

In addition, when the target object is located anywhere other than the center or top-right, the policy fails. In the OOD-position experiment 3, we relocated the target objects to a place other than the centre or the top-right corner. As expected, all VLAs failed. The end effector still travelled to the original location of the target object, picked up whatever object happened to occupy that spot, and dropped it in the basket.

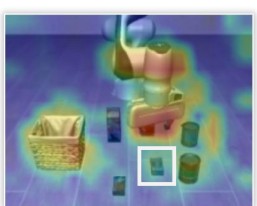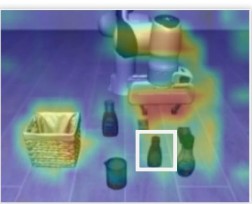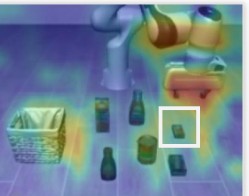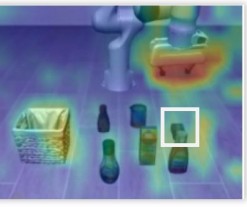

Figure 5: The object to grasp is framed in white. Even if the end effector is close to the object, the $\pi_0$ model is still not interested in the target object. Also, during the inference, $\pi_0$ always focuses on the posture of the end effector, suggesting that the image input is mainly for estimating the robot state.

As *text latent* encodes the task context, we can analyze which part of the image observation contributes most to the formation of *text latent* for $\pi_0$ with logit lens (nostalgebraist, 2020; Jiang et al., 2025). Fig. 5 highlights the information extracted from the image observation to build *text-latent* in 4 *libero-object* tasks at different timesteps. We find that the $\pi_0$ always absorbs the information about the destination and the robot pose to build *text latent*, with less attention on the object to grasp or treat all objects equally. Thus, $\pi_0$ relies on the text instruction instead of image information to differentiate the two grasping locations. This suggests a poor vision-language alignment and shortcut learning.

## 5 CONCLUSION

Existing VLAs show promising in-distribution generalization after fine-tuning, but their out-of-distribution (OOD) robustness is understudied due to a lack of suitable benchmarks. This is primarily because it is difficult to determine how far new tasks should deviate from the training data distribution. For instance, it would be unreasonable to expect a robot arm trained solely on tabletop manipulation to hold a steering wheel and drive a car. An ideal OOD benchmark should therefore comprise tasks that VLAs have the potential to complete but currently cannot. In this work, we find that by using the learned internal representation, $\pi_0$ can be augmented to solve novel tasks that it can not complete on its own. Since the injected knowledge is derived from the model's own learned representations, we conclude that $\pi_0$ inherently possesses the capability to complete these new tasks, although this ability appears to be latent or "locked". Tasks constructed in this manner are thus well-suited for our purpose, leading us to introduce an OOD benchmark named *libero-ood*. Besides $\pi_0$, we further evaluate other SOTA VLAs on LIBERO, including $\pi_0$-fast, UniVLA, and openvla-oft. Unfortunately, none of them achieved a success rate higher than 21%. Their failure may stem from the same issue as $\pi_0$, an inability to recombine learned representations into new skills, or skill representations don't emerge during the training. We thus make *libero-ood* public to encourage investigating their failure mode and developing more generalist models without using inference tricks like TLI or prompt switching.

**Reproducibility statement.** The code to reproduce the results of this work is anonymously at `https://anonymous.4open.science/r/iclr26_anonymous-3D4E/README.md`. The full videos of $\pi_0$-TLI on the proposed *libero-ood* benchmark can be found in the supplementary materials.

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

APPENDIX

# A    DISCLOSE OF LLM USAGE

The LLM is only used to polish the writing, mainly the introduction and conclusion.

# B    LIBERO-BENCHMARK

The *libero-goal* suite evaluates goal-completion capabilities, reflecting procedural knowledge (knowing how to complete a task). Conversely, the *libero-object* and *libero-spatial* suite assess object recognition, localization, and interaction capabilities, highlighting declarative knowledge (understanding entities and concepts). Tasks in *libero-object* share the same goal, picking up a specific object and placing it into a basket, but differ in object layouts. In contrast, *libero-goal* tasks require policies to achieve various goals within the same scene layout. The *libero-spatial* instead operates in different layouts to test whether the policy can find the correct bowl to pick and place it on the plate.

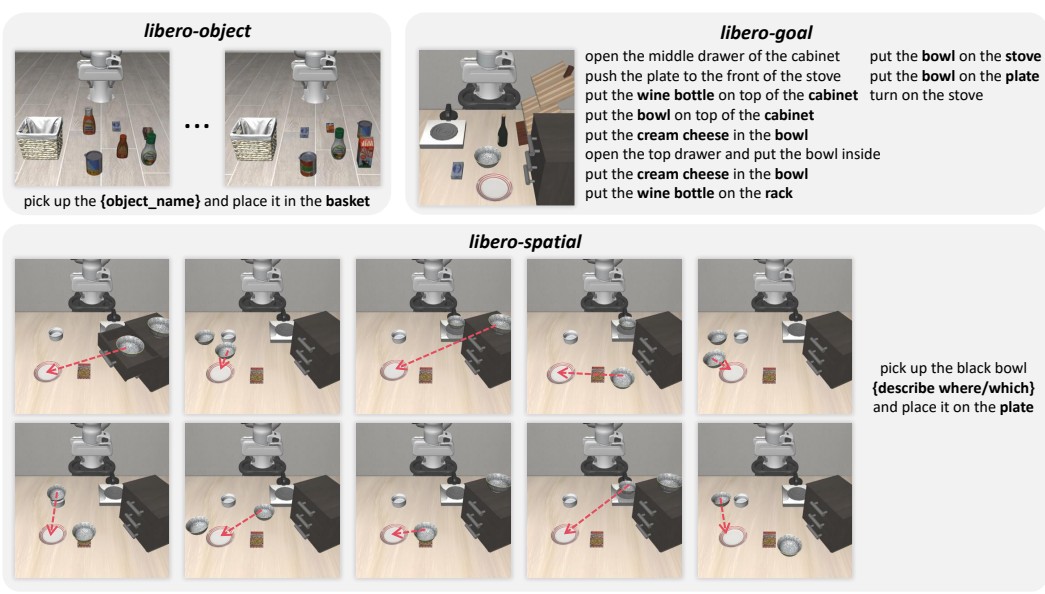

Figure 6:  The three basic libero task suites used to build extrapolated tasks. The *libero-object* tasks ask the robot to pick a specific object and place it in the basket; The *libero-goal* tasks operate in the same scene and require the robot to finish several different tasks where 7 of them are pick and place tasks and will be used to finish extrapolated tasks; The *libero-spatial* aims to test whether the robots can understand the space, and thus asks the robot to pick the specified bowl and place it on the plate.

## C    UNEMBEDDED PROMPTS

For *libero-goal*, we use $\mathcal{T}_1$ to get the alternative prompt, while for *libero-object*, we use $\mathcal{T}_2$.

| libero-object | libero-goal |
|---|---|
| ['ᴅ₤ 'ᴅ₤ '\U000e0062', 'ᴅ₤ ' sauce', 'QSize', 'QSize', 'ᴅ₤ '\U000e0062', '\U000e0062', 'QSize'] | ['\U000e0062', '\U000e0062', ' wine', ' bottle', '\U000e0062', 'Top', 'QSize', '\U000e0062', ' cabinet'] |
| ['ᴅ₤ 'ᴅ₤ '\U000e0062', 'QSize', ' soup', 'QSize', 'ᴅ₤ '\U000e0062', '\U000e0062', 'QSize'] | ['\U000e0062', '\U000e0062', 'ᴅ₤ '\U000e0062', '\U000e0062', ' plate'] |
| ['ᴅ₤ 'ᴅ₤ '\U000e0062', ' milk', 'QSize', 'QSize', 'ᴅ₤ '\U000e0062', '\U000e0062', 'QSize'] | ['\U000e0062', '\U000e0062', 'ᴅ₤ '\U000e0062', '\U000e0062', ' stove'] |
| ['ᴅ₤ 'ᴅ₤ '\U000e0062', ' salad', ' dressing', 'QSize', 'QSize', 'ᴅ₤ '\U000e0062', '\U000e0062', 'QSize'] | ['\U000e0062', '\U000e0062', ' cream', ' cheese', '\U000e0062', '\U000e0062', 'ᴅ₤] |
| ['ᴅ₤ 'ᴅ₤ '\U000e0062', 'ᴅ₤ 'QSize', 'QSize', 'ᴅ₤ '\U000e0062', '\U000e0062', 'QSize'] | ['Open', '\U000e0062', ' middle', 'ValueError', 'QSize', '\U000e0062', ' cabinet'] |
| ['ᴅ₤ 'ᴅ₤ '\U000e0062', ' cream', 'ᴅ₤ 'QSize', 'QSize', 'ᴅ₤ '\U000e0062', '\U000e0062', 'QSize'] | ['push', '\U000e0062', ' plate', 'QSize', '\U000e0062', ' front', 'QSize', '\U000e0062', ' stove'] |
| ['ᴅ₤ 'ᴅ₤ '\U000e0062', ' orange', 'ᴅ₤ 'QSize', 'QSize', ']==", '\U000e0062', '\U000e0062', 'QSize'] | ['Turn', 'QSize', '\U000e0062', ' stove'] |
| ['ᴅ₤ 'ᴅ₤ '\U000e0062', 'ᴅ₤ 'ᴅ₤ 'QSize', 'QSize', 'ᴅ₤ '\U000e0062', '\U000e0062', 'QSize'] | ['Open', '\U000e0062', 'Top', 'Einf', 'QSize', 'Einf', '\U000e0062', 'ᴅ₤ ' inside'] |
| ['ᴅ₤ 'ᴅ₤ '\U000e0062', 'ᴅ₤ 'QSize', 'QSize', 'ᴅ₤ '\U000e0062', '\U000e0062', ' basket'] | ['\U000e0062', '\U000e0062', ' wine', ' bottle', '\U000e0062', '\U000e0062', ' rack'] |
| ['ᴅ₤ 'ᴅ₤ '\U000e0062', 'QSize', ' sauce', 'QSize', 'QSize', 'ᴅ₤ '\U000e0062', '\U000e0062', 'QSize'] | ['\U000e0062', '\U000e0062', 'ᴅ₤ '\U000e0062', 'Top', 'QSize', '\U000e0062', ' cabinet'] |

Table 4: The alternative prompts for *libero-goal* and *libero-object*. It is hard to understand them, even if one knows the original prompts shown in Appendix B,

| Original Prompt | Prompt umembedded from *text-latent* (tokens) |
|---|---|
| pick up the black bowl next to the plate and place it on the plate | 'ᴅ₤ 'QSize', '\U000e0062', ' black', 'ᴅ₤ 'ᴅ₤ 'QSize', '\U000e0062', ' plate', 'QSize', 'QSize', 'QSize', 'QSize', '\U000e0062', 'QSize' |
| pick up the black bowl on the cookie box and place it on the plate | 'ᴅ₤ 'ᴅ₤ '\U000e0062', ' black', 'ᴅ₤ '\U000e0062', '\U000e0062', ' cookie', 'QSize', 'QSize', 'QSize', 'ᴅ₤ '\U000e0062', '\U000e0062', 'QSize' |
| pick up the black bowl on the wooden cabinet and place it on the plate | 'ᴅ₤ 'ᴅ₤ '\U000e0062', ' black', 'ᴅ₤ '\U000e0062', '\U000e0062', ' wooden', ' cabina', 'QSize', 'QSize', 'ᴅ₤ '\U000e0062', '\U000e0062', 'QSize' |
| pick up the black bowl on the ramekin and place it on the plate | 'ᴅ₤ 'ᴅ₤ '\U000e0062', ' black', 'ᴅ₤ '\U000e0062', '\U000e0062', 'ᴅ₤ 'kin', 'QSize', 'QSize', 'ᴅ₤ '\U000e0062', '\U000e0062', 'QSize' |
| pick up the black bowl on the stove and place it on the plate | 'ᴅ₤ 'ᴅ₤ '\U000e0062', ' black', 'ᴅ₤ '\U000e0062', '\U000e0062', ' stove', 'QSize', 'QSize', 'ᴅ₤ '\U000e0062', '\U000e0062', 'QSize' |
| pick up the black bowl next to the ramekin and place it on the plate | 'ᴅ₤ 'QSize', '\U000e0062', ' black', 'ᴅ₤ 'ᴅ₤ 'QSize', '\U000e0062', 'ᴅ₤ 'kin', 'QSize', 'QSize', 'QSize', '\U000e0062', '\U000e0062', 'QSize' |
| pick up the black bowl from table center and place it on the plate | 'ᴅ₤ 'QSize', '\U000e0062', ' black', 'QSize', ' FROM', ' table', 'ᴅ₤ 'QSize', 'QSize', 'ᴅ₤ '\U000e0062', '\U000e0062', 'ᴅ₤ |
| pick up the black bowl between the plate and the ramekin and place it on the plate | 'ᴅ₤ 'ᴅ₤ '\U000e0062', ' black', 'ᴅ₤ ' between', '\U000e0062', 'ᴅ₤ 'QSize', '\U000e0062', 'ᴅ₤ 'kin', 'QSize', 'QSize', 'ᴅ₤ '\U000e0062', '\U000e0062', 'QSize' |
| pick up the black bowl in the top drawer of the wooden cabinet and place it on the plate | 'QSize', 'QSize', '\U000e0062', ' black', 'QSize', '\U000e0062', '\U000e0062', 'Top', 'ᴅ₤ 'QSize', '\U000e0062', ' wooden', ' cabina', 'QSize', 'QSize', 'QSize', '\U000e0062', '\U000e0062', 'QSize' |
| pick up the black bowl next to the cookie box and place it on the plate | 'ᴅ₤ 'QSize', '\U000e0062', ' black', 'QSize', 'QSize', 'QSize', '\U000e0062', ' cookie', 'QSize', 'QSize', 'QSize', 'QSize', 'QSize', '\U000e0062', 'QSize' |

Table 5: Prompts decoded from $\mathcal{T}_3$ for *libero-spatial* tasks and can instruct $\pi_0$ to finish tasks with around 70% success rate. This enables adversarial attacks and private instructions.

# D BEHAVIOR VISUALIZATION

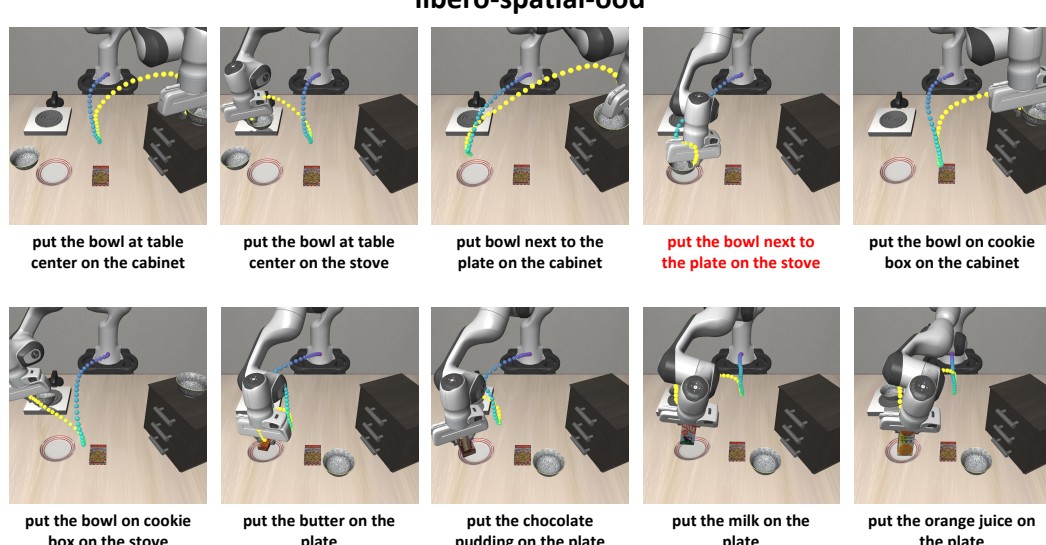

Figure 7: Trajectories of finishing tasks in *libero-ood*. The only failed task is highlighted in red. A fun fact is that these colored dots indicating the trajectory can be observed by the $\pi_0$ as well, while it is still robust to this visual perturbation. The only failure case is *put the bowl next to the plate on the stove*. It keeps picking up the bowl next to the plate and placing it on the plate. We suspect that the second task context is not strong enough to trigger the behavior "put on the stove", or the context of the first task is too strong to be fully removed from the residual stream, so the policy keeps implementing the behaviors of the first task and put the bowl on the plate.

# E  DETAIL TASK PERFORMANCE

libero-goal-ood

| | Put The Bbq Source On The Plate | Put The Cream Cheese In The Basket | Put The Cream Cheese On The Plate | Put The Cream Cheese On The Stove | Put The Cream Cheese On Top Of The Cabinet | Put The Orange Juice On The Stove | Put The Tomato Sauce On Top Of The Cabinet | Put The Wine Bottle In The Bowl | Put The Wine Bottle On The Plate | Put The Wine Bottle On The Stove | Total Success Rate (%) |
|---|---|---|---|---|---|---|---|---|---|---|---|
| openvla-oft | 0.0 | 0.0 | 0.0 | 0.0 | 0.0 | 0.0 | 0.0 | 0.0 | 0.0 | 0.0 | 0.0 |
| $\pi_0$-fast | 0.1 | 0.0 | 1.0 | 0.0 | 0.2 | 0.0 | 0.0 | 0.0 | 0.1 | 0.0 | 0.14 |
| $\pi_0$ | 0.0 | 0.1 | 0.1 | 0.0 | 0.0 | 0.0 | 0.0 | 0.0 | 0.0 | 0.0 | 0.02 |
| $\pi_0$-TEI | 0.9 | 0.8 | 0.2 | 0.5 | 0.3 | 0.7 | 0.6 | 0.0 | 0.1 | 0.1 | 0.42 |
| $\pi_0$-TLI | 1.0 | 0.9 | 1.0 | 1.0 | 1.0 | 0.7 | 1.0 | 0.5 | 0.8 | 0.6 | 0.85 |
| $\pi_0$-TEI-TLI | 0.9 | 0.8 | 1.0 | 1.0 | 0.9 | 0.8 | 0.9 | 0.6 | 0.7 | 0.9 | 0.85 |
| $\pi_0$-TLI* | 0.0 | 0.0 | 0.2 | 0.4 | 1.0 | 0.3 | 0.6 | 0.2 | 0.5 | 0.1 | 0.33 |

libero-spatial-ood

| | Put The Bowl At Table Center On The Cabinet | Put The Bowl At Table Center On The Stove | Put The Bowl Next To The Plate On The Cabinet | Put The Bowl Next To The Plate On The Stove | Put The Bowl On Cookie Box On The Cabinet | Put The Bowl On Cookie Box On The Stove | Put The Butter On The Plate | Put The Chocolate Pudding On The Plate | Put The Milk On The Plate | Put The Orange Juice On The Plate | Total Success Rate (%) |
|---|---|---|---|---|---|---|---|---|---|---|---|
| openvla-oft | 0.0 | 0.0 | 0.0 | 0.0 | 0.0 | 0.0 | 0.0 | 0.0 | 0.0 | 0.0 | 0.0 |
| $\pi_0$-fast | 0.0 | 0.1 | 0.0 | 0.0 | 0.0 | 0.0 | 0.0 | 1.0 | 0.4 | 0.2 | 0.17 |
| $\pi_0$ | 0.0 | 0.1 | 0.0 | 0.0 | 0.0 | 0.0 | 0.6 | 0.7 | 0.2 | 0.0 | 0.16 |
| $\pi_0$-TEI | 0.1 | 0.6 | 0.0 | 0.1 | 0.0 | 0.1 | 0.5 | 1.0 | 0.9 | 0.8 | 0.41 |
| $\pi_0$-TLI | 0.7 | 1.0 | 0.8 | 0.0 | 1.0 | 0.8 | 1.0 | 0.9 | 1.0 | 0.9 | 0.81 |
| $\pi_0$-TEI-TLI | 0.4 | 0.9 | 0.4 | 0.0 | 0.4 | 0.9 | 0.9 | 1.0 | 1.0 | 1.0 | 0.69 |
| $\pi_0$-TLI* | 0.0 | 0.4 | 0.2 | 0.0 | 0.2 | 0.2 | 0.2 | 0.5 | 0.1 | 0.0 | 0.18 |

