# OpenReview forum: "VLAs Are Confined yet Capable of Generalizing to Novel Tasks"
_ICLR.cc/2026/Conference — ICLR 2026 Conference Withdrawn Submission_

### Official Review · Reviewer_KYX7 · 2025-10-31

**Soundness:** 2
**Presentation:** 4
**Contribution:** 2
**Rating:** 4
**Confidence:** 4

**Summary:**

This paper provides an insightful diagnosis of why Vision-Language-Action (VLA) models fail at compositional generalization. It demonstrates that SOTA models (like $\pi_0$) do learn the necessary sub-skills but cannot autonomously compose them for novel tasks. Using a novel inference-time technique called Text Latent Interpolation (TLI), the authors prove this latent capability exists (boosting success from 9% to 83%) and identify Spatial Overfitting—where the model memorizes locations instead of understanding objects—as a root cause for this failure.

**Strengths:**

1. Addresses the critical and well-defined problem of compositional generalization in robotics.
2. Offers a deep diagnosis of why SOTA models fail, rather than just proposing a new model.
3. The TLI method is a clever experimental tool that compellingly proves the model's "locked" potential (9% $\rightarrow$ 83% success).
4. Contributes a valuable and challenging new benchmark (libero-ood) that effectively exposes this common VLA failure mode.
5. Identifies "Spatial Overfitting" as a specific root cause, revealing models are memorizing locations, not learning object semantics.

**Weaknesses:**

1. The proposed TLI method is an inference-time "trick," not a fundamental solution. The model itself remains incapable of autonomous composition.
2. TLI is impractical for real-world use, as it requires a priori knowledge of the task decomposition (e.g., knowing "A $\rightarrow$ C" = "A $\rightarrow$ B" + "B $\rightarrow$ C").
3. The method relies on manually-tuned hyperparameters (like interpolation speed), limiting its generality.
4. All experiments are in simulation, making it unclear if the observed flaws are fundamental or an artifact of the training environment.

**Questions:**

See weakness.

**Details Of Ethics Concerns:**

Nope.

---

### Official Review · Reviewer_7KJb · 2025-10-31

**Soundness:** 2
**Presentation:** 2
**Contribution:** 2
**Rating:** 2
**Confidence:** 5

**Summary:**

This paper investigates skill composability in VLAs via text latent interpolation and introduces an OOD benchmark, revealing limited generalization despite fine-tuning.

**Strengths:**

The proposed method demonstrates that latent skill representations can be combined to solve novel tasks, improving OOD performance significantly.

**Weaknesses:**

1. The motivation of this paper is unclear in the abstract. The example given, "VLAs might successfully put the cream cheese in the bowl and put the bowl on top of the cabinet, yet still fail to put the cream cheese on top of the cabinet," is incorrect. Why? Existing VLA models can do this; why can't they?
2. The abstract's conclusion, "Further analysis reveals that VLAs share a common pattern to exhibit spatial overfitting, associating object names with where the object is spatially located in the demonstrated scene rather than achieving true object and goal understanding," is also problematic. Many papers have already proposed and proven that large models only possess associative abilities but lack logical reasoning capabilities. This paper's innovation is weak.

**Questions:**

1. The motivation of this paper is unclear in the abstract. The example given, "VLAs might successfully put the cream cheese in the bowl and put the bowl on top of the cabinet, yet still fail to put the cream cheese on top of the cabinet," is incorrect. Why? Existing VLA models can do this; why can't they?
2. The abstract's conclusion, "Further analysis reveals that VLAs share a common pattern to exhibit data overfitting, associating object names with where the object is spatially located in the demonstrated scene rather than achieving true object and goal understanding," is also problematic. Many papers have already proposed and proven that large models only possess associative abilities but lack logical reasoning capabilities. This paper's innovation is weak.
3. How was the libero-ood benchmark validated to ensure it fairly assesses generalization?
4. Were hyperparameters like λ tuned systematically, and how do they impact the robustness of TLI across tasks?
5. What measures were taken to ensure that the compared VLAs were evaluated under identical conditions and fine-tuning protocols?
6. Does the VLA model truly have an OOD problem? If the amount of data is large enough, does it still have an OOD problem?

---

### Official Review · Reviewer_hfJt · 2025-11-02

**Soundness:** 3
**Presentation:** 3
**Contribution:** 3
**Rating:** 4
**Confidence:** 4

**Summary:**

The paper introduces a mechanistic interpretability approach to probe what VLAs have learned internally. The authors also release LIBERO-OOD as a new benchmark to test compositional generalization in VLAs. It demonstrates large (9×) improvement using a simple, interpretable mechanism

However, there are also some concerns and limtations about the experiments. First and also most important: All experiments are in simulation (LIBERO) — no real-robot validation. The VLA model are usurally pretrained on real world data. The LIBERO itself lack of realistic rendering, making spatial undersanding and reasoning harder than real. The benchmark might not be fair. Also TLI is manua: human selects which tasks to blend; not an autonomous reasoning process.

**Strengths:**

1. First to identify and manipulate task-specific text latents inside VLAs.
2. Demonstrates large (9×) improvement using a simple, interpretable mechanism.
3. Introduces LIBERO-OOD, a principled OOD suite targeting skill recomposition.
4. Reveals potential for “encrypted prompts” and backdoor control via latent injection.

Overall the idea and method are interesting and might be useful to understand VLA models better and help improve VLA models.

**Weaknesses:**

1. All experiments are in simulation (LIBERO) and no real-robot validation. The VLA model are usurally pretrained on real world data. The LIBERO itself lack of realistic rendering, making spatial undersanding and reasoning harder than real. The benchmark might not be fair.
2. TLI is manual: human selects which tasks to blend; not an autonomous reasoning process.
3. Due to the scope of LIBERO, the paper focused mainly on pick-and-place tasks — limited coverage of complex manipulation dynamics.

**Questions:**

See weakness above.

---

### Note · Authors · 2025-11-24

**Comment:**

We thank reviewers and their efforts for evaluating the paper.

**Withdrawal Confirmation:**

I have read and agree with the venue's withdrawal policy on behalf of myself and my co-authors.